# Optimizing Prompts for Text-to-Image Generation

**Yaru Hao**,* **Zewen Chi**,* **Li Dong, Furu Wei**
Microsoft Research
`https://github.com/microsoft/LMOps`

## Abstract

Well-designed prompts can guide text-to-image models to generate amazing images. However, the performant prompts are often model-specific and misaligned with user input. Instead of laborious human engineering, we propose prompt adaptation, a general framework that automatically adapts original user input to model-preferred prompts. Specifically, we first perform supervised fine-tuning with a pretrained language model on a small collection of manually engineered prompts. Then we use reinforcement learning to explore better prompts. We define a reward function that encourages the policy to generate more aesthetically pleasing images while preserving the original user intentions. Experimental results on Stable Diffusion show that our method outperforms manual prompt engineering in terms of both automatic metrics and human preference ratings. Moreover, reinforcement learning further boosts performance, especially on out-of-domain prompts. The pretrained checkpoints are available at `https://aka.ms/promptist`. The demo can be found at `https://aka.ms/promptist-demo`.

## 1 Introduction

Generative foundation models can be prompted to follow user instructions, including language models [Brown et al., 2020, Chowdhery et al., 2022, Smith et al., 2022], and text-to-image models [Ramesh et al., 2021a, 2022, Saharia et al., 2022, Rombach et al., 2022]. It has been recognized that prompt design plays an essential role in the generation quality. We need to adjust the prompt to make the model better understand our intentions and produce higher-quality results [Reynolds and McDonell, 2021, Zhou et al., 2022b]. The problem is severe in text-to-image models because the capacity of their text encoders, such as CLIP text encoder [Radford et al., 2021] in Stable Diffusion [Rombach et al., 2022], is relatively small. Empirical observations also confirm that common user input is often insufficient to produce aesthetically pleasing images with current models.

Prior efforts implement manual prompt engineering towards specific text-to-image models [Liu and Chilton, 2021, Oppenlaender, 2022, Parsons, 2022], typically adding some modifiers to the original input. However, it is laborious and sometimes infeasible to conduct manual prompt engineering. Besides, the manually engineered prompts often cannot be transferred between various model versions. Therefore, it is necessary to find a systematic way to automatically align user intentions and various model-preferred prompts.

In this work, we propose a prompt adaptation framework for automatic prompt engineering via reinforcement learning. Specifically, we first perform supervised fine-tuning with a pretrained language model (e.g., GPT) on a small collection of manually engineered prompts. The finetuned model is used to initialize the prompt policy network for reinforcement learning. Next, the model is trained by exploring optimized prompts of user inputs, where diverse beam search [Vijayakumar et al., 2016] is used to ensure generation quality and diversity. The training objective is to maximize

---

* Equal contribution.

37th Conference on Neural Information Processing Systems (NeurIPS 2023).

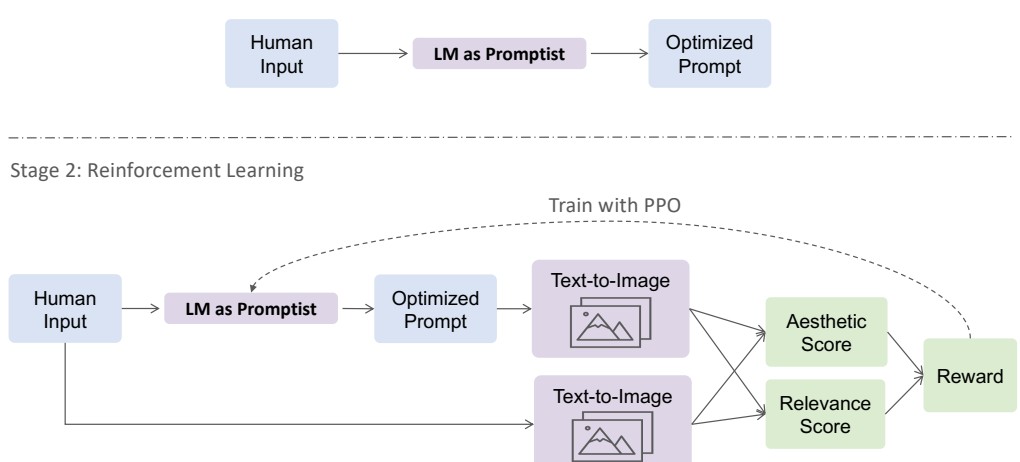

Figure 1: Overview of PROMPTIST training: (1) supervised fine-tuning (SFT) on manually engineered prompts; (2) reinforcement learning (RL) to increase the rewards of generated images after prompt optimization.

the reward, which is defined as a combination of relevance scores and aesthetic scores of generated images. The relevance score reflects how much the original user intentions are retained after prompt adaptation. The aesthetic score indicates what degree the generated images are aesthetically pleasing.

We conduct experiments with the publicly available Stable Diffusion models [Rombach et al., 2022]. We evaluate our method using both the automatic reward metric and human preference ratings. Experimental results show that the optimized prompts outperform human-engineered ones and the original inputs. Human preference ratings also show consistent improvements across in-domain and out-of-domain prompts. Moreover, we find that reinforcement learning is more favorable than supervised fine-tuning, especially on out-of-domain user inputs. Overall, we show that language models can serve as a prompt interface that optimizes user input into model-preferred prompts.

Our contributions are as follows:

- We propose a general prompt optimization framework that adapts user input to model-preferred prompts.

- We collect user queries and conduct extensive experiments on text-to-image generation.

- Experimental results show that our method outperforms manual prompt engineering in terms of both automatic metrics and human preference ratings.

## 2   Methods

The goal of our prompt adaptation framework is to automatically perform prompt engineering. Given user input of the text-to-image generator, our model learns to generate model-preferred prompts that obtain better output images while preserving their original intentions. Figure 1 presents the overview of our method. The prompt optimization model is named PROMPTIST, which is built upon a pretrained language model, such as GPT [Brown et al., 2020]. We first collect a set of human-engineered examples and use them to conduct supervised fine-tuning (Section 2.1). Next, we perform reinforcement learning (Section 2.3) to maximize the target reward (Section 2.2), which improves both relevance and quality of generated images.

## 2.1 Supervised fine-tuning

Initialized with a pretrained generative language model, the policy model is first finetuned on a set of prompt pairs before reinforcement learning. A parallel prompt corpus $\mathcal{D} = \{(\boldsymbol{x}, \boldsymbol{y})\}$ contains prompt pairs of original user inputs $\boldsymbol{x}$ and manually engineered examples $\boldsymbol{y}$. The training objective is to maximize the log-likelihood with teacher forcing:

$$\mathcal{L}_{\text{SFT}} = -\mathbb{E}_{(\boldsymbol{x},\boldsymbol{y})\sim\mathcal{D}} \log p(\boldsymbol{y}|\boldsymbol{x}) \tag{1}$$

where the finetuned weights are used to initialize the policy network in reinforcement learning.

**Collect human demonstrations**  We collect human-engineered prompts from Lexica[2]. Most prompts are composed of two parts, i.e., main content that describes the user's intention, and some modifiers that customize the art style, such as artist names, and popular elements. We use the crawled human-engineered prompts as targets. In order to obtain parallel data, we use three methods to construct their source inputs. First, we extract the main contents by trimming the modifiers and regard them as original user inputs. Second, we randomly remove or shuffle some modifiers and use the remaining texts as source inputs. Third, we use the OpenAI API `text-davinci-002` to rephrase the main contents and the human-engineered prompts, respectively. We find that the template "`[Input] Rephrase:`" works well in practice and translates input to a more user-friendly version. As shown in Table 1, given a target prompt "`dieselpunk blue wolf with fuzzy tail, concept art, dramatic, fantasy, pixiv`", there are four source prompts collected.

Table 1: An example of a human-engineered prompt and four types of our constructed source prompts.

| | |
|---|---|
| **Human-engineered target prompt** | dieselpunk blue wolf with fuzzy tail, concept art, dramatic, fantasy, pixiv |
| **Main content** | dieselpunk blue wolf with fuzzy tail |
| **Main content with random modifiers** | dieselpunk blue wolf with fuzzy tail, dramatic |
| **Rephrasing of main content** | A blue wolf with a fuzzy tail that looks like it belongs in a dieselpunk setting. |
| **Rephrasing of target prompt** | This is a dieselpunk-style blue wolf with a fuzzy tail. It looks like it could be from a fantasy or dramatic piece of artwork. |

## 2.2 Reward definition

We measure the quality of optimized prompts from two aspects, namely relevance and aesthetics. The goal motivates us to define the reward function $\mathcal{R}(\cdot)$ from the above two perspectives.

First, we measure whether the generated images are relevant to the original input prompt after prompt adaptation. To be specific, we first sample images by the text-to-image model conditioned on the optimized prompt, respectively. Then, we compute CLIP [Radford et al., 2021] similarity scores to measure how relevant the generated images and the original input prompts are. The resulting relevance score is defined as:

$$f_{\text{rel}}(\boldsymbol{x},\boldsymbol{y}) = \mathbb{E}_{i_{\boldsymbol{y}}\sim\mathcal{G}(\boldsymbol{y})}[f_{\text{rel}}(\boldsymbol{x}, i_{\boldsymbol{y}})] \tag{2}$$

$$f_{\text{rel}}(\boldsymbol{x}, i_{\boldsymbol{y}}) = min(20 * g_{\text{CLIP}}(\boldsymbol{x}, i_{\boldsymbol{y}}) - 5.6, 0) \tag{3}$$

where $i_{\boldsymbol{y}} \sim \mathcal{G}(\boldsymbol{y})$ means sampling images $i_{\boldsymbol{y}}$ from the text-to-image model $\mathcal{G}$ with $\boldsymbol{y}$ as input prompt, and $g_{\text{CLIP}}(\cdot, \cdot)$ stands for the CLIP similarity function. Notice that we always compute the similarity between the generated images and the original input prompt, which ensures the relevance score reflects the user preferences. We determine the specific form of the relevance score according to the approximate range of the clip score. Experiments show that this form works well in reinforcement learning. If the relevance score is relatively reasonable (larger than 0.28), we encourage the model to generate more aesthetically pleasing images.

Second, we employ the aesthetic predictor[3] to quantify aesthetic preferences. The predictor builds a linear estimator on top of a frozen CLIP model, which is trained by human ratings in the Aesthetic

---

[2]`https://lexica.art`
[3]`https://github.com/christophschuhmann/improved-aesthetic-predictor`

Visual Analysis [Murray et al., 2012] dataset. The aesthetic score is defined as:

$$f_{\text{aes}}(\boldsymbol{x}, \boldsymbol{y}) = \mathbb{E}_{i_{\boldsymbol{x}} \sim \mathcal{G}(\boldsymbol{x}), i_{\boldsymbol{y}} \sim \mathcal{G}(\boldsymbol{y})} [g_{\text{aes}}(i_{\boldsymbol{y}}) - g_{\text{aes}}(i_{\boldsymbol{x}})] \tag{4}$$

where $g_{\text{aes}}(\cdot)$ denotes the aesthetic predictor, and $i_{\boldsymbol{y}}, i_{\boldsymbol{x}}$ are the images generated by the prompts $\boldsymbol{y}$ and $\boldsymbol{x}$, respectively. Notice that both $g_{\text{CLIP}}(\cdot)$ and $g_{\text{aes}}(\cdot)$ require the CLIP model, so we can share the CLIP forward pass during reward computation.

Finally, we define the overall reward by combining the above scores with an additional KL penalty, which is between the policy model $\pi_{\boldsymbol{\theta}}$ and the supervised finetuned model $\pi_{\text{SFT}}$ with coefficient $\eta$:

$$\begin{aligned} \mathcal{R}(\boldsymbol{x}, \boldsymbol{y}) = {} & f_{\text{aes}}(\boldsymbol{x}, \boldsymbol{y}) + f_{\text{rel}}(\boldsymbol{x}, \boldsymbol{y}) \\ & - \eta \log \frac{\pi_{\boldsymbol{\theta}}(\boldsymbol{y}|\boldsymbol{x})}{\pi_{\text{SFT}}(\boldsymbol{y}|\boldsymbol{x})} \end{aligned} \tag{5}$$

The KL term is added to mitigate the overoptimization issue [Ouyang et al., 2022].

## 2.3 Reinforcement learning

Starting from the supervised fine-tuning, we further finetune our model with reinforcement learning. We employ proximal policy optimization (PPO) [Schulman et al., 2017], which is empirically data-efficient and of reliable performance. As a text generation problem, prompt optimization can be viewed as a Markov decision process (MDP) $\langle \mathcal{S}, \mathcal{A}, r, f_{\text{st}}, \gamma \rangle$ with a finite state space $\mathcal{S}$, action space $\mathcal{A}$, reward function $r$, state-transition probability function $f_{\text{st}}$, and a discount term $\gamma$. In an episode of prompt adaptation, the initial state $\boldsymbol{x} \in \mathcal{S}$ is the input prompt with $n$ tokens $\boldsymbol{x} = (x_1, \ldots, x_n)$ where each token $x$ is from a finite vocabulary $\mathcal{V}$. At $t$-th time step, the agent selects an action $y_t \in \mathcal{V}$ according to the current policy model $y_t \sim \pi(y|\boldsymbol{x}, \boldsymbol{y}_{<t})$. With a deterministic state transition, the next state is $(\boldsymbol{x}, \boldsymbol{y}_{<t+1}) = (x_1, \ldots, x_n, y_1, \ldots, y_t)$. The episode ends when the agent selects an end-of-sentence action. The goal of the agent is to maximize the accumulated expected reward $\mathbb{E}_{\boldsymbol{x}, \boldsymbol{y}} \sum_t \gamma^t r(\boldsymbol{x}, \boldsymbol{y}_{<t}) = \mathbb{E}_{\boldsymbol{x}, \boldsymbol{y}} \mathcal{R}(\boldsymbol{x}, \boldsymbol{y})$.

Let $\pi_{\boldsymbol{\theta}}$ denote the policy model to be trained, we maximize the accumulated expected reward over a training set $\mathcal{D}' = \{\boldsymbol{x}\}$:

$$\mathcal{J} = \mathbb{E}_{\boldsymbol{x} \sim \mathcal{D}', \boldsymbol{y} \sim \pi_{\boldsymbol{\theta}}} [\mathcal{R}(\boldsymbol{x}, \boldsymbol{y})] \tag{6}$$

We implement both the policy model $\pi_{\boldsymbol{\theta}}$ and the value function model as generative language models, with the language modeling head and the regression head, respectively. The parameters of the two models are initialized from the supervised finetuned policy model $\pi_{\text{SFT}}$ and are optimized during reinforcement learning. The supervised finetuned model $\pi_{\text{SFT}}$ and the score function model are frozen during training. Besides, we employ the clipped probability ratios [Schulman et al., 2017] to avoid large policy updates.

# 3 Experiments

We conduct experiments on public text-to-image model Stable Diffusion v1.4[4] and v1.5[5] . We use the DPM solver [Lu et al., 2022] to accelerate image sampling and set the denoising steps to 20.

## 3.1 Data collection

For supervised fine-tuning, we collect 90k target prompts from Lexica website and construct four types of source prompts as described in Section 2.1, obtaining 360k paired data in total. At the reinforcement learning stage, we only require source prompts and the policy can explore better rephrasings itself. We use three types of data: (1) in-domain prompts from DiffusionDB [Wang et al., 2022], which is a gallery of prompts specified by real users. We use the user input (main content) for exploration and the manually engineered prompt (with modifiers) for comparison, (2) out-of-domain image captions from COCO dataset [Chen et al., 2015], (3) image labels from ImageNet-21k [Deng et al., 2009], the sizes of which are 600k, 600k and 40k respectively. We empirically observe that

---

[4]`https://huggingface.co/CompVis/stable-diffusion-v1-4`
[5]`https://huggingface.co/runwayml/stable-diffusion-v1-5`

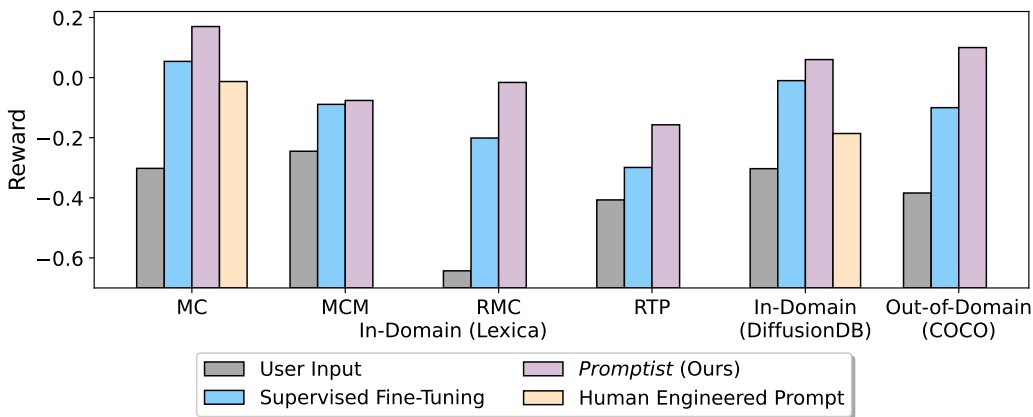

Figure 2: Reward comparison of optimized prompts with other baselines on in-domain and out-of-domain data. For in-domain Lexica prompts, we evaluate on four augmentations: main content (MC), main content with random modifiers (MCM), rephrasing of main content (RMC) and rephrasing of target prompt (RTP). Results indicate that the text-to-image model benefits a lot from our method.

Table 2: Absolute reward improvements of supervised fine-tuning and reinforcement learning. It is observed that RL generally outperforms the SFT-only model.

|  | In-Domain (Lexica) | | | | In-Domain | Out-of-Domain |
|  | MC | MCM | RMC | RTP | (DiffusionDB) | (COCO) |
| --- | --- | --- | --- | --- | --- | --- |
| SFT | 0.36 | 0.16 | 0.44 | 0.11 | 0.29 | 0.28 |
| RL | 0.47 | 0.17 | 0.63 | 0.25 | 0.36 | 0.48 |
| Gain | **+31%** | **+6%** | **+43%** | **+127%** | **+24%** | **+71%** |

human-engineered prompts from Lexica perform better than those from DiffusionDB so we use the former in supervised fine-tuning. To improve the data diversity, we add image caption data and image label data in reinforcement learning. To avoid model bias in certain data formats, we randomize the capitalization of the first letter of each prompt and randomly add periods at the end of it.

## 3.2 Settings

For the policy model, we use GPT-2 [Radford et al., 2019] with 117M parameters, which is a multi-layer Transformer [Vaswani et al., 2017] decoder pretrained with causal language modeling.

**Supervised fine-tuning**    We finetune GPT-2 to predict the target prompt conditioned on the source prompt with teacher forcing. The input format is `[Source] Rephrase:[Target]`. We use a batch size of 256, a learning rate of 5e-5, and a max length of 512. We finetune the model for 15k steps and choose a slightly underfitting checkpoint according to the validation loss which aims to avoid overfitting and provide a proper exploration space for the policy.

**Reinforcement learning**    We train the policy with Proximal Policy Optimization [Schulman et al., 2017, PPO]. The value and policy network are initialized from the supervised finetuned model. The parameters of the value function are separated from the policy to avoid excessive competition between two objectives. To guarantee the quality and diversity of exploration, we adopt diverse beam search [Vijayakumar et al., 2016] with a beam size of 8 and a diversity penalty of 1.0. We find that having too long rephrasings occasionally produces aesthetically pleasing but misleading results, especially for short user input like image labels. In order to prevent the model from only exploring long completions, the maximum generation length at each step is set to a random value from 15 to 75 so that the policy can learn to adjust the generation length for each prompt. We randomly choose one of the returned completions after diverse beam search to update the policy. We generate three images per prompt and compute the average reward to reduce variance. We train the policy for 12k episodes, four PPO epochs per batch with one minibatch each, with a batch size of 256 and a constant

learning rate of 5e-5. The value loss coefficient and the KL reward coefficient are kept at 2.3 and 0.2 respectively. We do not cherry-pick checkpoints and directly use the final checkpoint for evaluation. Please refer to the Appendix A for more training details and Appendix B for computational resources.

**Evaluation**   In order to evaluate how text-to-image models benefit from the prompt adaptation, we compare the reward value computed by two automatic predictors (Section 2.2). Moreover, we use human preference ratings to demonstrate real user feedback. We adopt beam search with a beam size of 8 and a length penalty of 1.0. We evaluate our method on held-out data from training distribution, including in-domain data from Lexica with four augmentations, in-domain data from DiffusionDB, and out-of-domain COCO data. Each category contains 256 prompts. In-domain data has corresponding manually engineered prompts for comparison, and the out-of-domain data is used to verify whether our method can generalize to new domains. Except for the user input and manually engineered baseline, we also consider the supervised finetuned model as a baseline that can reflect the importance of reinforcement learning.

## 3.3   Results

**Optimized prompts obtain higher reward improvements than manual engineering.**   We evaluate optimized prompts on held-out data by generating three images for each prompt and computing the average reward value. Figure 2 shows that the reward value can be improved regardless of the engineering method, which suggests the misalignment problem between user-friendly prompts and model-preferred prompts is serious. Compared with the strong baseline of manually engineered prompts, optimized prompts can still achieve considerable reward improvements. Furthermore, optimized prompts perform even better on rephrased versions (i.e., RMC, and RTP), and out-of-domain data. These prompts are more user-friendly but cause more significant reward drops on generation results, especially on the rephrasing of the main content. Benefiting from automatic prompt engineering, optimized prompts can align well between two different domains from users and text-to-image models respectively.

We also present the evaluation results of the aesthetic score and relevance score respectively in Table 4. We empirically found that the generated images are relevant enough to the input prompt if the relevance score (CLIP score) is around 0.26. As mentioned at Section 2.2, we design the reward function which encourages the model to generate more aesthetically pleasing images if the relevance score is good enough. On the DiffusionDB dataset, our RL method improves the SFT baseline in terms of relevance score from 0.25 to 0.26, and the human-engineered baseline also obtains a relevance score of 0.26. Moreover, the aesthetic score of our model is improved significantly over both the human-engineered prompts and the supervised fine-tuned model. It demonstrates that our method generates images with good relevance and much better aesthetic scores.

We provide some images generated by user input and its corresponding optimized prompt in Table 3. Each group consists of three images generated by different random seeds. We observe that images generated by user input are intuitively uninspiring while optimized prompts can not only retain the original intentions but also induce the model to produce more remarkable results. For example, generated images are crude when prompted with "A rabbit is wearing a space suit". After prompt optimization, generated images become more bright and more expressive.

**Reinforcement learning can further boost the reward value.**   Reinforcement learning in our method is supposed to perform better on out-of-domain data through explorations. To quantify its effect, we compute the ratio of reward improvements after fine-tuning and reinforcement learning. As shown in Table 2, reinforcement learning brings 31%, 24%, and 71% average improvements on in-domain main content from Lexica, DiffusionDB, and out-of-domain COCO data. In-domain prompts are very similar to the data we used in supervised fine-tuning, so reward improvements are relatively saturated in the first stage and improvements of reinforcement learning on them are correspondingly smaller. Oppositely, out-of-domain data such as COCO captions are more similar to user input and unseen during the first stage. The policy must learn to adapt better to new domains through exploration, so their improvements on these prompts are more prominent. Surprisingly, although in-domain Lexica prompts and their augmentations are not used, reinforcement learning still exhibits better generalization capability on them. The boost is remarkable on those prompts that fine-tuning cannot optimize well (43% on rephrasings of main content and 127% on rephrasings of

Table 3: Images generated by user input and optimized prompts using Stable Diffusion v1.4. Each group contains three images generated with three different random seeds. We observe that optimized prompts can generate more aesthetically pleasing images than original user input.

| User Input | Optimized Prompt |
| --- | --- |
| A rabbit is wearing a space suit | A rabbit is wearing a space suit, digital Art, Greg rutkowski, Trending cinematographic artstation |
|  |  |
| Several railroad tracks with one train passing by | several railroad tracks with one train passing by, hyperdetailed, artstation, cgsociety, 8 k |
|  |  |
| The roof is wet from the rain | the roof is wet from the rain, intricate, elegant, highly detailed, digital painting, artstation, concept art, smooth, sharp focus, illustration, |
|  |  |
| Cats dancing in a space club | Cats dancing in a space club, digital painting, artstation, concept art, soft light, hdri, smooth, sharp focus, illustration, fantasy, |
|  |  |

target prompt). These results suggest that given appropriate human queries, reinforcement learning can optimize them to adapt to different domains and boost reward improvements.

To further demonstrate the effectiveness of our framework, we also present the results of our model on Stable Diffusion v1.5 in Appendix C, comparisons with the heuristic baseline in Appendix D and the results on different categories and lengths of prompts in Appendix E.

## 3.4 Human evaluation

The reward function of our model is defined by two automatic metrics, aesthetic score and relevance score predicted by neural networks, which may have some discrepancies from real human feedback. Therefore, we additionally evaluate whether optimized prompts actually make humans more satisfied. We generate two images for each user input and the optimized prompt. Afterward, three held-out annotators are asked to rank the two groups of images in preference order and we compute the average preference distribution. Results are shown in Table 5. We observe that annotators generally prefer images generated by optimized prompts over their original input. Compared with manually

Table 4: Evaluation of the aesthetic score and relevance score on DiffusionDB.

|  | Aesthetic | Relevance |
|---|---|---|
| User Input | 5.47 | 0.28 |
| Human Engineered Prompt | 5.87 | **0.26** |
| Supervised Fine-tuning | 6.15 | 0.25 |
| PROMPTIST(Ours) | **6.26** | **0.26** |

Table 5: Human evaluation results. The different colors represent how many images generated by corresponding prompts are considered more aesthetically pleasing. The orange block means that both prompts produce equally pleasing images.

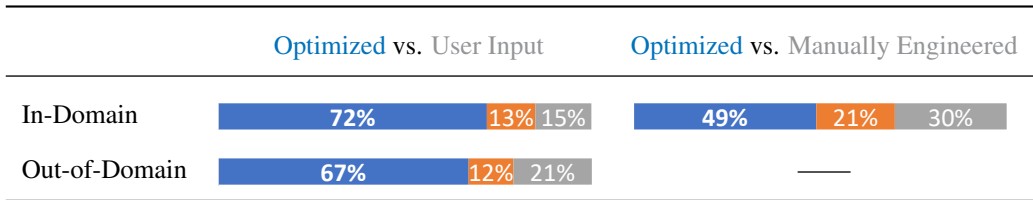

engineered prompts, optimized prompts yield less gain over user input. It suggests that the aesthetic score can measure the quality of generated images to some extent, it would be better if direct human feedback is included in the reward function.

### 3.5 Ablation of source prompt augmentation

As described in Section 2.1, we crawl human-engineered prompts as target prompts and use the main content without any modifiers as source prompts. To enable the supervised fine-tuned model to generalize better on unseen domains, we propose different augmentation methods that improve the diversity of source prompts. We compare the fine-tuning performance with and without the augmentation strategy and results are shown in Figure 3. We observe that fine-tuning with source prompt augmentation brings consistent improvements on both in-domain held-out data and out-of-domain data. From the results on MCM, adding some random modifiers to the user input slightly obtains reward improvement but it is not as distinct as the improvement brought by fine-tuning, indicating that we should customize modifiers for each individual prompt and automatic prompt engineering is a promising way to tackle it. Compared with other data, prompts rephrased by `text-davinci-002` are more difficult to optimize at the fine-tuning stage and they benefit more from reinforcement learning. Overall, source prompt augmentation makes the fine-tuned model generalize better and is important in our prompt adaptation framework.

## 4   Related work

**Prompt engineering.**   Manual prompt engineering is a natural way to optimize prompts. Manually designed cloze-style prompts have been used to probe knowledge from pre-trained language models [Petroni et al., 2019, Dai et al., 2022]. In addition to knowledge probing, models are also prompted to handle NLP tasks with manually designed prefix prompts [Brown et al., 2020, Du et al., 2021]. Recent work has explored how to write prompts to improve performance [Wei et al., 2022]. Despite the success of manually-crafted prompts, designing prompts takes time and experience [Shin et al., 2021] and can be sub-optimal [Jiang et al., 2020]. In particular, when using text-to-image models, users have to carefully select and compose sentences to achieve a certain visual style [Liu and Chilton, 2021, Oppenlaender, 2022, Parsons, 2022]. Thus, various methods focus on automatically searching prompts by mining [Jiang et al., 2020], paraphrasing [Haviv et al., 2021], and text generation [Gao et al., 2021]. Besides, continuous prompt methods treat the prompts as additional continuous parameters of pre-trained models and directly optimize the parameters on downstream tasks [Li and Liang, 2021, Tsimpoukelli et al., 2021, Zhou et al., 2022a]. However, continuous prompt methods require access to manipulating the model, and the learned prompts lack interpretability. In contrast,

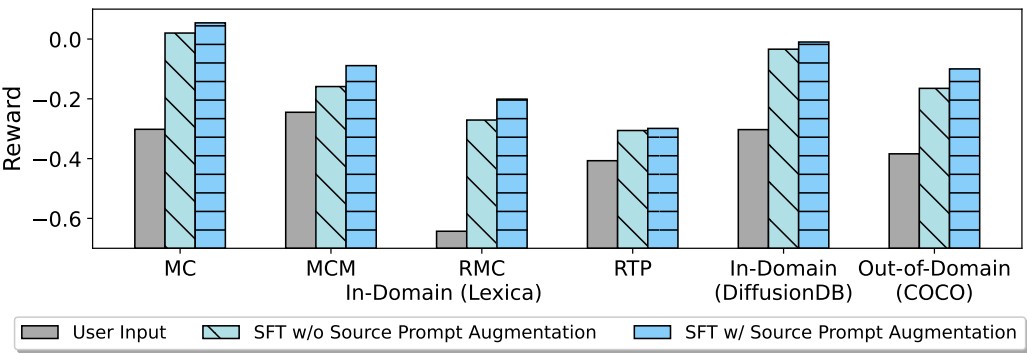

Figure 3: Reward comparison of supervised fine-tuning with or without source prompt augmentation. For in-domain Lexica prompts, we evaluate on four augmentations: main content (MC), main content with random modifiers (MCM), rephrasing of main content (RMC) and rephrasing of target prompt (RTP). It is observed that source prompt augmentation in supervised fine-tuning can boost performance on both in-domain data and out-of-domain data.

our methods directly optimize prompts in text format, which can fit in black-box downstream systems such as text-to-image models.

**Learning from human feedback.** Our work is related to research on learning from human feedback, which has been widely studied in machine learning problems. Several studies propose to continually improve dialogue systems by collecting human feedback after deployment [Hancock et al., 2019, Shuster et al., 2020, Xu et al., 2022]. Besides, human feedback has also been also applied to human-in-the-loop methods for entity linking [Klie et al., 2020], semantic parsing [Yao et al., 2019], etc. Recent research on reinforcement learning from human feedback (RLHF) has shown promising results on machine learning problems, ranging from classical RL tasks [Christiano et al., 2017, Ibarz et al., 2018] to a wide range of natural language processing tasks, including text summarization [Stiennon et al., 2020, Ziegler et al., 2019], dialogue [Jaques et al., 2019], and general text generation tasks [Ouyang et al., 2022]. Differently, our goal is to automatically optimize prompts for text-to-image models.

**Text-to-image models.** Text-to-image synthesis models are typically trained to generate images conditioned on text. Text-to-image synthesis has been widely studied using GANs [Reed et al., 2016a,b, Tao et al., 2022]. More recently, text-to-image models are further improved with large-scale auto-regressive models [Ramesh et al., 2021b, Ding et al., 2021] or diffusion-based models [Rombach et al., 2022, Gu et al., 2022].

## 5    Conclusion

We propose to automatically optimize prompts for text-to-image models so that the user input and model-preferred prompts can be well aligned. We evaluate our method with Stable Diffusion. Experimental results show that prompt adaptation outperforms human prompt engineering and supervised fine-tuning, in terms of automatic metrics and human evaluation. The exploration nature of reinforcement learning enables the model to go beyond teacher forcing, which improves generalization over out-of-domain examples. The proposed method is flexible to align human intentions and model-favored languages. Although our experiments are conducted on text-to-image models, the framework can be easily applied to other tasks for prompt adaptation. Rather than using automatic score functions as rewards, we can directly use human feedback as supervision to train a reward model [Ouyang et al., 2022]. Moreover, using a larger-size language model as the prompt interface tends to improve the optimization quality.

## 6    Limitations

We crawl human-engineered prompts from the Lexica website as golden prompts to guide the supervised fine-tuning process. The crawled prompts contain some biases. For example, we observe

that they tend to generate more artwork instead of realistic photographs because most of them contain one or more artist names. Besides, the proportion of prompts about portraits is relatively higher than those about other categories. Although the reinforcement learning stage can mitigate these issues, it would be better to balance the art styles and objects at the beginning. Moreover, we currently only apply our framework to text-to-image models. As the proposed framework is general to prompt-guided generation, we will apply it to other generative models like text-only models and text-to-video models for future work.

## Acknowledgments

We would like to thank Tan Yan for the helpful discussions.

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

# Appendix

## A   Hyperparameter settings

Table 6: Hyperparameter settings of supervised fine-tuning (SFT) and reinforcement learning (RL).

| Hyperparameters | SFT | RL |
|---|---|---|
| Batch Size | 256 | 256 |
| Learning Rate | 5e-5 | 5e-5 |
| Training Steps | 15000 | 12000 |
| Max Length | 512 | 512 |
| Dropout | 0.0 | 0.0 |
| Optimizer | Adam | Adam |
| Adam $\epsilon$ | 1e-6 | 1e-6 |
| Adam $\beta$ | (0.9, 0.999) | (0.9, 0.95) |
| Weight Decay | 0.1 | 1e-6 |

## B   Computational budget

Our experiments are implemented on V100 (32GB) GPU.

Table 7: Computational budget of supervised fine-tuning (SFT) and reinforcement learning (RL).

| | SFT | RL |
|---|---|---|
| The Number of GPUs | 4 | 32 |
| GPU Hours | 3 hours | 2.5 days |

## C   Results on Stable Diffusion v1.5.

Table 8: Results on Stable Diffusion v1.5

| | Lexica | DiffusionDB | COCO |
|---|---|---|---|
| User Input | -0.31 | -0.32 | -0.37 |
| Human Engineered Prompt | -0.05 | -0.18 | - |
| Supervised Fine-tuning | -0.04 | -0.16 | -0.1 |
| PROMPTIST(Ours) | **0.05** | **0.06** | **0.11** |

# D   Comparisons with heuristic baseline

Table 9: Combinations of common tags.

| Tag | Content |
|-----|---------|
| 1 | artstation, highly detailed, elegant |
| 2 | 8 k, trending on artstation, concept art |
| 3 | digital painting, intricate, fantasy |
| 4 | illustration, smooth, octane render |
| 5 | digital art, 8k, intricate |
| 6 | highly detailed, elegant, smooth |

Table 10: Comparisons with the heuristic baseline.

| Data | User | Tag1 | Tag2 | Tag3 | Tag4 | Tag5 | Tag6 | Human | SFT | Ours |
|------|------|------|------|------|------|------|------|-------|-----|------|
| Lexica | -0.32 | 0.07 | -0.06 | 0.06 | -0.17 | -0.05 | -0.28 | -0.02 | 0.03 | **0.14** |
| DiffusionDB | -0.3 | 0 | -0.07 | -0.16 | -0.1 | -0.17 | -0.31 | -0.21 | -0.01 | **0.06** |
| COCO | -0.38 | -0.24 | -0.29 | -0.2 | -0.33 | -0.32 | -0.41 | - | -0.1 | **0.1** |

To compare the performance of our proposed framework with the heuristic baseline, we select the top 15 frequent tags from human-engineered prompts and randomly combine them to create six groups of common tags. The specific tags are presented in Table 9. We concatenate the user input with these common tags and compute their reward. The results are in Table 10.

While using these common tags can improve the reward to some extent, we found that their performance varies significantly across different domains. For instance, tag3 performs well on COCO and Lexica but poorly on DiffusionDB. It suggests that relying on a handful of common hand-selected tags may not be practical in real-world scenarios. In contrast, our proposed framework can perform well across domains and improve a lot over the common tags.

# E Results on different categories and lengths of prompts

We aim to validate the effectiveness of our method on different categories and different lengths. In Figure 2, we divide the prompts into several categories according to the prompt pattern, there are MC, MCM, RMC, RTP, in-domain DiffusionDB and out-of-domain COCO. Results show that optimized prompts are generally effective for all these categories. For semantic categories, these prompts have no clear boundaries. Therefore, we use RoBERTa-Large Liu et al. [2019] to get the sentence embedding of each prompt and perform K-means clustering on these prompts and divide them into five categories. We list their proportion and their reward in Table. For length ablation, we also classify them into five categories according to the length of input tokens. The results are in Table 12.

We observe that the performance of different lengths and semantic categories varies slightly but our model can improve the reward generally. When conducting reinforcement learning, we build large-scale prompts from both in-domain data and out-of-domain data, which cover a wide range of prompts with different lengths and semantic categories.

Table 11: Results on different semantic categories of prompts.

| Cluster | 1 | 2 | 3 | 4 | 5 |
|---|---|---|---|---|---|
| Proportion | 0.18 | 0.08 | 0.17 | 0.21 | 0.36 |
| User Input | -0.39 | -0.29 | -0.37 | -0.34 | -0.31 |
| PROMPTIST (Ours) | -0.01 | 0.04 | 0.13 | 0.06 | 0.1 |

Table 12: Results on different lengths of prompts.

| Length | 0~10 | 10~20 | 20~30 | 30~40 | >40 |
|---|---|---|---|---|---|
| Proportion | 0.21 | 0.48 | 0.2 | 0.07 | 0.04 |
| User Input | -0.48 | -0.33 | -0.18 | -0.28 | -0.22 |
| PROMPTIST (Ours) | -0.02 | 0.11 | 0.08 | 0.05 | 0.06 |

