# OpenReview forum: "Optimizing Prompts for Text-to-Image Generation"
_NeurIPS.cc/2023/Conference — NeurIPS 2023 spotlight_

### Official Review · Reviewer_51Hz · 2023-07-03

**Soundness:** 3 good
**Presentation:** 4 excellent
**Contribution:** 4 excellent
**Rating:** 7
**Confidence:** 4

**Summary:**

The author proposes a new general framework to optimize prompts for text-to-image generation. Specifically, the framework goes through two stages: a supervised fine-tuning stage on a small sets of human engineered prompts and a reinforcement learning process to explore better prompts.

The supervised data were web-crawled from Lexica and is augmented with different methods for diversity. During the reinforcement learning process, the model is optimized with two losses. A relevance loss that is obtained with a frozen CLIP encoder to determine if the image obtained from the modified prompt matches with the original text description; an aesthetic loss that is evaluated against an aesthetic predictor trained on human ratings.

Overall, the paper is technically sound and introduces a new framework that can be easily adapted to use.

**Strengths:**

The paper is nicely written and easy to follow.

The introduced a two-stage pipeline that trains a LM to modify human prompts for better text-to-image generation. The qualitative and quantitative results are solid.

**Weaknesses:**

The experiment section could use more work to further validate the robustness of the general framework.

The choice of the model architecture for image generation (Stable Diffusion V1.4) and the LM (GPT-2) seem a bit arbitrary, especially given that these model were iterated at a very fast pace. Given new iteration on the development of image generation model, it could very well be the case that the limitations are addressed with more advanced image generation model.

No experiments were done on the different choices of model architecture.

**Questions:**

In equation (4), the aesthetic score is the averaged difference in the aesthetic predictor score of the modified prompt and the original prompt. I wonder why the image generation from the original prompt is used.
What are the expected behaviour when using different choices of image generation model or language model?

Could a smaller LM reach the same level of performance / generated quality with a larger image generation model?

**Limitations:**

As already identified by the authors, the generated results are highly biased towards artistic generation as opposed to photo-realistic images. It would be better if the promptist model could handle a wide range of art styles in generated images. The problem could be mitigated when bigger and better image generation model or language model are used.

---

> ### Author Rebuttal · Authors · 2023-08-10
>
> *Q1: Why is the image generated from the original prompt used in the aesthetic score?*
>
> A1: There are two possible reasons for the high aesthetic score of an optimized prompt. One is the optimized prompt has high aesthetic improvements over the original prompt, the other is the original prompt itself has a high aesthetic score. Intuitively, the former should have a higher reward than the latter. Therefore, we use the aesthetic improvement over the original prompt instead of the absolute aesthetic score as the reward.
> \
> \
> *Q2: No experiments were done on the different choices of model architecture. What are the expected behaviour when using different choices of image generation model or language model?*
>
> A2: (1) Different image generation model. We also conduct experiments on Stable Diffusion v1.5. The results are as follows.
>
> |     Data           |     User input    |     Human    |     SFT      |     Promptist    |
> |--------------------|-------------------|--------------|--------------|------------------|
> |     COCO           |     -0.37         |     N/A      |     -0.1     |     0.11         |
> |     DiffusionDB    |     -0.32         |     -0.18    |     -0.16    |     0.06         |
> |     Lexica         |     -0.31         |     -0.05    |     -0.04    |     0.05         |
>
> (2) There are other more capable language models like LLAMA and MPT that can potentially yield better results. However, using these models will bring a higher training cost.
>
> In general, using more capable text-to-image models and language models can potentially improve the generation quality further. The proposed framework is general to various choices of models and we will explore the use of these models in the future.
> \
> \
> *Q3: Could a smaller LM reach the same level of performance / generated quality with a larger image generation model?*
>
> A3: Compared with smaller image generation models, the larger models usually have better baseline generation quality and are relatively more robust to prompt patterns. Therefore, we can use a relatively smaller LM to optimize prompts.
> \
> \
> *Q4: Given new iteration on the development of image generation model, it could very well be the case that the limitations are addressed with more advanced image generation model.*
>
> A4: It is because of the fast iteration of models that we require an automatic framework to optimize prompts. The old optimized prompts could be less effective on the new models and it is laborious to conduct human-engineering for all versions of models. The proposed framework can generally optimize prompts automatically for them through reinforcement learning. Furthermore, more high-quality training data and larger model scale can improve the generation quality of text-to-image models, but there is still a gap between model-preferred prompts and human-preferred prompts. It is challenging for current text-to-image models (890M for SD) to achieve the strong enough capability of text understanding and be totally robust to the prompt change. Therefore, the framework is effective because better prompts can improve the generation quality further.

---

> > ### Comment · Reviewer_51Hz · 2023-08-17
> >
> > Thank the author for the response!
> >
> > The response answers my concern and I maintain my initial evaluation.

---

### Official Review · Reviewer_Kg4J · 2023-07-03

**Soundness:** 3 good
**Presentation:** 3 good
**Contribution:** 3 good
**Rating:** 6
**Confidence:** 5

**Summary:**

The paper presents a novel framework for automatically optimizing prompts in text-to-image generation models. The authors combine supervised fine-tuning with a pretrained language model and reinforcement learning to explore better prompts. The goal is to generate more aesthetically pleasing images while preserving the original user intentions. Experimental results show that the proposed method outperforms manual prompt engineering in terms of both automatic metrics and human preference ratings. This work contributes significantly to the field of text-to-image generation by providing a systematic way to align user intentions and model-preferred prompts.

**Strengths:**

1. **Originality**: The paper presents a novel framework, "Prompt Adaptation", for automatically optimizing prompts in text-to-image generation models. This approach is innovative as it combines supervised fine-tuning with a pretrained language model and reinforcement learning to explore better prompts. This is a significant contribution to the field of text-to-image generation.
2. **Quality**: The paper is well-structured and provides a clear and detailed description of the proposed framework. The authors have also provided a comprehensive analysis of the results, demonstrating that their method outperforms manual prompt engineering in terms of both automatic metrics and human preference ratings.
3. **Clarity**: The paper is well-written and easy to understand. The authors have done a good job of explaining complex concepts in a clear and concise manner.
4. **Significance**: The proposed framework has the potential to significantly improve the quality of images generated from text prompts, making it a valuable tool for a wide range of applications in areas such as digital art, advertising, and content creation.

**Weaknesses:**

1. **Data Bias**: The authors acknowledge that the prompts used for supervised fine-tuning, which were crawled from the Lexica website, contain biases. For example, they tend to generate more artwork instead of realistic photographs and have a higher proportion of prompts about portraits than other categories. This could limit the generalizability of the proposed framework.
2. **Limited Application**: Currently, the proposed framework is only applied to text-to-image models. While the authors mention plans to extend it to other generative models like text-only models and text-to-video models, this limitation could restrict the current applicability of the framework.

**Questions:**

1. How does the proposed framework handle complex prompts that require the generation of images with multiple objects or scenes?
2. Could the authors provide more details on how the reinforcement learning stage mitigates the biases in the prompts used for supervised fine-tuning?

**Limitations:**

The main limitation of this study is the bias in the prompts used for supervised fine-tuning. The authors have acknowledged this and suggested that balancing the art styles and objects at the beginning could help address this issue. However, they do not provide a clear strategy for achieving this balance.

---

> ### Author Rebuttal · Authors · 2023-08-10
>
> *Q1: Prompts crawled from Lexica website contain data bias. This could limit the generalizability of the proposed framework.*
>
> A1: There are several approaches to mitigate the data bias such as data cleaning, semantic deduplication and directly writing high-quality prompt pairs. However, these approaches require much human labor cost. We leave utilizing these data cleaning methods for future work.
> \
> \
> *Q2: Limited Application: Currently, the proposed framework is only applied to text-to-image models.*
>
> A2: The misalignment of human prompts and model-preferred prompts is more severe in text-to-image models because the capacity of their text encoders is limited. Thus, we currently apply the method to text-to-image models. The proposed framework is general to other models of which different prompts can affect the generation quality. We leave them for future work.
> \
> \
> *Q3: How does the proposed framework handle complex prompts that require the generation of images with multiple objects or scenes?*
>
> A3: The training data we used is large and contains those prompts with many tokens (75 at most for Stable Diffusion), especially in rephrasing augmented prompts and prompts from DiffusionDB. They ensure that the framework can handle complex prompts and generate accurate and high-quality images.
> \
> \
> *Q4: Could the authors provide more details on how the reinforcement learning stage mitigates the biases in the prompts used for supervised fine-tuning?*
>
> A4: For supervised fine-tuning, the training data is the human-engineered prompts crawled from the website, which have some bias. When conducting reinforcement learning, we only need the source prompt, so we use many out-of-domain data which has nearly no bias compared to the supervised data. Moreover, we also use diverse beam search for prompt exploration in reinforcement learning. It means that the optimized prompts we explore are diverse every time. In summary, the first strategy can mitigate the bias of source prompt distribution and the second strategy can mitigate the bias of modifier distribution.
> \
> Additionally, the talk (https://eecs.berkeley.edu/research/colloquium/230419) illustrates the benefits of reinforcement learning over supervised fine-tuning, which is also applicable to our work.

---

> > ### Comment · Reviewer_Kg4J · 2023-08-16
> >
> > I acknowledge that I have read the response and I keep my original score.

---

### Official Review · Reviewer_brRy · 2023-07-06

**Soundness:** 2 fair
**Presentation:** 4 excellent
**Contribution:** 2 fair
**Rating:** 6
**Confidence:** 4

**Summary:**

The paper proposes Promptist a prompt engine that leverages a pre-trained LLM (specifically GPT-2) to learn to generate prompts preferred by the model based on original prompt from the user. The approach is simple and very effective.

The policy network is first fine-tuned with supervision based on pairs of user inputs and manually engineered prompts, obtained from Lexica.art (90k manually engineered prompts). Synthetic prompts were generated using few different approaches, and then paired with the original engineered prompt from Lexica.art to create paired samples. This initial policy network is used as part of the reward function along with aesthetic and relevance scores. Prompts generated by Promptist performs even better than heavily manually engineered prompts. Experiments section also includes human evaluation comparing against main prompt and manually engineered prompt. Detailed experiments section provides useful insights.


**Strengths:**

S1) Clarity of writing

S2) Detailed experiments section providing useful insights

S3) The proposed approach produces better results (aesthetics) than heavily manually engineered prompts

S4) Simple and very effective approach


**Weaknesses:**

W1) Generalization to other architectures is not clear. Results are shown only for SD 1.4.

W2) Generalization to the LM in Promptist is also not clear. Current experiments use GPT-2

W3) CLIPScore of 0.28 seems to be a magic number. See Line 91 and equation 3. It is not clear how to adapt this to new data sets

W4) Effectiveness of Promptist for different categories and lengths of prompts is also not clear


**Questions:**

Q1) Would the prompts for the training set need to come from prompts used with SD 1.4+ family? Perhaps just requires the T2I model to be trained based on LAION-2B-en subset?

**Limitations:**

Yes

---

> ### Author Rebuttal · Authors · 2023-08-10
>
> *Q1: Generalization to other architectures. (Results are shown only for SD 1.4)*
>
> A1: We also conduct experiments on Stable Diffusion v1.5. The results are as follows.
>
> |     Data           |     User input    |     Human    |     SFT      |     Promptist    |
> |--------------------|-------------------|--------------|--------------|------------------|
> |     COCO           |     -0.37         |     N/A      |     -0.1     |     0.11         |
> |     DiffusionDB    |     -0.32         |     -0.18    |     -0.16    |     0.06         |
> |     Lexica         |     -0.31         |     -0.05    |     -0.04    |     0.05         |
>
> The results show that the optimized prompts are effective on SD 1.5 as well, i.e., consistently outperform human-engineered prompts and supervised-fine-tuning. Our framework is not limited to a specific text-to-image model version and the training data is designed independently of the model version. Therefore, our approach can easily generalize to other text-to-image models.
> \
> \
> *Q2: Generalization to other LMs. (Current experiments use GPT-2)*
>
> A2: Our proposed framework is agnostic to the choice of language model, and we use GPT-2 in our experiments to verify the effectiveness of optimized prompts. There are other more capable language models like LLAMA and MPT that can potentially yield better results. However, using these models will bring a higher training cost. We plan to explore the use of these language models in the future.
> \
> In summary, the proposed framework is general and can be applied to various text-to-image models and language models.
> \
> \
> *Q3: CLIPScore of 0.28 seems to be a magic number. It is not clear how to adapt this to new data sets.*
>
> A3: We studied the CLIP scores of generated images with the input text and found that the generated images are highly relevant to the input text when the CLIP scores are greater than 0.28. Therefore, we designed the reward function with a binary reward threshold at 0.28. We can directly use it on new datasets, because the number 0.28 depends on which CLIP model we choose.
> \
> \
> *Q4: Effectiveness of Promptist for different categories and lengths of prompts is also not clear.*
>
> A4: We divide the prompts into several categories according to the prompt pattern (Figure 2), there are MC, MCM, RMC, RTP, in-domain DiffusionDB and out-of-domain COCO. Results show that optimized prompts are generally effective for all these categories. For semantic categories, these prompts have no clear boundaries. Therefore, we use RoBERTa-Large to get the sentence embedding of each prompt and perform K-means clustering on these prompts and divide them into five categories. We list their proportion and their reward in the following table.
>
> |     Cluster       |     1        |     2        |     3        |     4        |     5        |
> |-------------------|--------------|--------------|--------------|--------------|--------------|
> |     Proportion    |     0.18     |     0.08     |     0.17     |     0.21     |     0.36     |
> |     User Input    |     -0.39    |     -0.29    |     -0.37    |     -0.34    |     -0.31    |
> |     Promptist     |     -0.01    |     0.04     |     0.13     |     0.06     |     0.1      |
>
> For length ablation, we also classify them into 5 categories according to the length of input tokens. The results are as follows.
>
> |     Length        |     0~10     |     10~20    |     20~30    |     30~40    |     >40      |
> |-------------------|--------------|--------------|--------------|--------------|--------------|
> |     Proportion    |     0.21     |     0.48     |     0.2      |     0.07     |     0.04     |
> |     User Input    |     -0.48    |     -0.33    |     -0.18    |     -0.28    |     -0.22    |
> |     Promptist     |     -0.02    |     0.11     |     0.08     |     0.05     |     0.06     |
>
> We observe that the performance of different lengths and semantic categories varies slightly but our model can improve the reward generally. When conducting reinforcement learning, we build large-scale prompts from both in-domain data and out-of-domain data, which cover a wide range of prompts with different lengths and semantic categories.
> \
> \
> *Q5: Would the prompts for the training set need to come from prompts used with SD 1.4+ family? Perhaps just requires the T2I model to be trained based on LAION-2B-en subset?*
>
> A5: The human-engineered prompts are collected from the Lexica website, where the prompts are designed for not only SD 1.4 but also many other text-to-image models. The proposed framework focuses on adapting user prompts to model-preferred ones. It does not matter whether the text-to-image model is trained on LAION-2B or not.

---

> > ### Comment · Reviewer_brRy · 2023-08-17
> >
> > Thanks to the authors for clarifying all of my questions/concerns.

---

### Official Review · Reviewer_zvr2 · 2023-07-07

**Soundness:** 2 fair
**Presentation:** 3 good
**Contribution:** 3 good
**Rating:** 6
**Confidence:** 4

**Summary:**

This paper proposes a method for improving user-provided prompts for text-to-image models to make them more intent-aligned and aesthetically appealing. They use the following method:
- Collect a dataset of paired basic and optimized user prompt.
- Finetune a LM to take in a basic prompt and produce an optimized prompt.
- Define a reward function based on CLIP text-image alignment and aesthetic appeal. (The aesthetic appeal is a new head trained on top of CLIP embeddings using a dataset of human judgements.)
- Use PPO with diverse beam search sampling to finetune the prompt-improvement LM to train against this reward.
They show that the optimized prompts score better than even human-engineered prompts on aesthetic quality according to human evaluations.

**Strengths:**

Originality: There is plenty of work on finetuning text-to-image models for text-image alignment and aesthetic scores, as well as on designing good hard prompts. I have not seen a paper which specifically optimizes prompts through RL.
Quality: The paper's experiments and evaluation seem thorough.
Clarity: the paper is well written.
Significance: It's a known issue that there are specific keywords which can significantly improve image quality and that casual users of text-to-image models might not know to use these. This seems like a meaningful use case.

**Weaknesses:**

- I wish there were comparisons to other methods of improving text-to-image alignment and aesthetics. The paper does not compare against alternate methods (such as learning soft prompts or finetuning the model directly), claiming these are not preferred because they require you to backprop through the model. They also yield less-interpretable outputs. These are reasonable points, so I think this is a minor weakness.
- The related work section also mentions methods for automatically searching for prompts. I'd either like to see comparisons so some of these methods or an explanation of why these comparisons aren't relevant.

- There is no analysis on how reliable the CLIP score and the aesthetic score are.  (Though the results are backed up by human evaluation.)


**Questions:**

* What does it mean "If the relevance score is relatively reasonable (larger than 0.28), we encourage the model to generate more aesthetically pleasing images".  Does this mean you use a binary reward thresholded at .28?
* In eqn 3, what are the constants 20 and -5.6 for? These seem arbitrary. (If it's just hyperparameters from a sweep, mention that.)

* I wish there a comparison with a heuristic baseline which just includes a handful of common hand-selected tags which can apply across most inputs (e.g. "artstation, hyperdetailed, concept art"). Basically, I'm wondering how well Promptist is able to target the generated tags to the particular user prompt or whether it's has mostly just learned a few common tags which produce aesthetic images of all kinds. Alternatively, a supplement with a couple pages of non-cherrypicked optimized prompts would also answer this for me.
* Relatedly, how does the diversity of the styles in Promptist prompts compare to the human-optimized prompts? (i.e. is it possible that human prompts are preferred less because humans ask for a more diverse set of image styles, not all of which score highly on the aesthetic score?)

**Limitations:**

The authors point out the issue that the training dataset is biased towards certain artistic styles. This analysis seems adequate.

---

> ### Author Rebuttal · Authors · 2023-08-10
>
> *Q1: Does this mean you use a binary reward thresholded at .28? In eq 3, what are the constants 20 and -5.6 for?*
>
> A1: Yes, the relevance reward has a binary threshold at 0.28. We empirically observe that the image and the text are sufficiently relevant when the CLIP score is 0.28. Eq3 can be also written as $20 * min(g_{clip} - 0.28, 0)$, where 20 is a hyperparameter to balance the scale of the relevance score and the aesthetic score. When the CLIP score is larger than 0.28, we do not consider its further gain and encourage the model to generate more aesthetic images. When it is smaller than 0.28, the reward will decrease with a slope of 20.
> We will state them in a clearer way in the camera-ready version.
> \
> \
> *Q2: I'd either like to see comparisons for automatically searching for prompts in the related work or an explanation of why these comparisons aren't relevant.*
>
> A2: Automatically searching for prompts [1,2,3] also aim to rewrite prompts to improve the downstream performance. [1] and [2] focus on knowledge extraction from BERT and [3] focus on few-shot learning of BERT. The main difference between our method and them is that they directly use downstream task labels as supervision to search for better prompts and cannot utilize the unsupervised data. They are more like the first stage of our framework, while our method can utilize the large-scale unsupervised data and the free-form exploration in reinforcement learning can improve further over supervised fine-tuning.
>
> [1] How Can We Know What Language Models Know? Jiang et al., 2020.
> \
> [2] BERTese: Learning to Speak to BERT. Haviv et al., 2021.
> \
> [3] Making Pre-trained Language Models Better Few-shot Learners. Gao et al., 2021.
> \
> \
> *Q3: Comparison with the heuristic baseline which just includes a handful of common hand-selected tags.*
>
> A3: To compare the performance of our proposed framework with the heuristic baseline, we select the top15 frequent tags from human-engineered prompts and randomly combine them to create six groups of common tags. The specific tags are as follows.
>
> |     tag    |     text                                          |
> |------------|---------------------------------------------------|
> |     1      |     artstation, highly detailed, elegant          |
> |     2      |     8 k, trending on artstation, concept   art    |
> |     3      |     digital painting, intricate, fantasy          |
> |     4      |     illustration, smooth, octane render           |
> |     5      |     digital art, 8k, intricate                    |
> |     6      |     highly detailed, elegant, smooth              |
>
> We concatenate the user input with these common tags and compute their reward. The results are as follows.
>
> |     Data           |     user input    |     tag1     |     tag2     |     tag3     |     tag4     |     tag5     |     tag6     |     Human    |     SFT      |     Promptist    |
> |--------------------|-------------------|--------------|--------------|--------------|--------------|--------------|--------------|--------------|--------------|------------------|
> |     COCO           |     -0.38         |     -0.24    |     -0.29    |     -0.2     |     -0.33    |     -0.32    |     -0.41    |     N/A      |     -0.1     |     0.1          |
> |     DiffusionDB    |     -0.3          |     0        |     -0.07    |     -0.16    |     -0.1     |     -0.17    |     -0.31    |     -0.21    |     -0.01    |     0.06         |
> |     Lexica         |     -0.32         |     0.07     |     -0.06    |     0.06     |     -0.17    |     -0.05    |     -0.28    |     -0.02    |     0.03     |     0.14         |
>
> While using these common tags can improve the reward to some extent, we found that their performance varies significantly across different domains. For instance, tag3 performs well on COCO and Lexica but poorly on DiffusionDB. It suggests that relying on a handful of common hand-selected tags may not be practical in real-world scenarios. In contrast, our proposed framework can perform well across domains and improve a lot over the common tags.
> \
> \
> *Q4: The diversity of the optimized prompts compared with the human-optimized prompts.*
>
> A4: We propose two strategies to promote diversity:
> \
> (1) (Line 72) When constructing data, we randomly remove or shuffle some modifiers of the human-engineered prompts as source prompt augmentations. It corresponds to the scenario that the human input contains specific styles based on their requirements. We encourage the model to maintain them while adding further improvements or choosing a better modifier order.
> \
> (2) (Line 150) We observe that both top-p sampling and beam search limit the exploration space make the optimization prompts less diverse. To prevent the model from learning a trivial solution of adding same modifiers for all prompts, we use diverse beam search to explore optimized prompts to ensure both the quality and the diversity.
> \
> \
> *Q5: There is no analysis on how reliable the CLIP score and the aesthetic score are.*
>
> A5: (1) CLIP score: Both TorchMetric (https://torchmetrics.readthedocs.io/en/stable/multimodal/clip_score.html) and HuggingFace (https://huggingface.co/docs/diffusers/main/en/conceptual/evaluation) state that “CLIP score was found to have high correlation with human judgement”.
> \
> (2) Aesthetic score: The aesthetic score predictor is trained on AVA [4] dataset, which contains over 250,000 images along with a rich variety of annotations. It has been widely used for aesthetic analysis.
> \
> Therefore, they are reliable for evaluation. We did not conduct quantitative reliability analysis because it requires much human effort. Moreover, our main goal is to validate the effectiveness of prompt optimization with a reasonable reward, so we directly use them for evaluation.
>
> [4] Ava: A large-scale database for aesthetic visual analysis. Murray et al., 2012.

---

> > ### Comment · Reviewer_zvr2 · 2023-08-16
> > **Concerns mostly addressed**
> >
> > Thanks for addressing my concerns in the "heuristic baseline" experiment. I will raise my score, though I still think the final version of the paper would benefit from a few additional changes:
> >
> > * In the appendix, include a few pages of non-cherrypicked human prompts and optimized prompts so it's easy to see the differences.
> > * Based on your Q2 response, I think comparisons to methods for automatically searching for prompts would be helpful, though not required.

---

### Decision · Program_Chairs · 2023-09-21

**Decision:**

Accept (spotlight)

**Comment:**

This paper proposes a prompt adaptation method to generate model-preferred prompts from original user inputs for text-to-image generation models. The proposed method first performs supervised fine-tuning with a pre-trained language model on manually engineered prompts, and then explores improved prompts via reinforcement learning with a reward function based on the aesthetic appeal and text-image alignment.

Initial concerns include the lack of comparisons with other methods, the reliability of the CLIP score and the aesthetic score, the generalization to other architecture and data, and limited application. Reviewers acknowledged that the rebuttal addressed most of these concerns. Finally, all reviewers gave positive final ratings. The AC agrees with the reviewers that the idea is novel, the proposed RL-based method is simple yet effective, and the paper is generally well-written. Therefore, the AC recommends accept. Reviewers did raise valuable concerns that should be addressed. The authors are encouraged to make the necessary changes in the camera-ready version.